# Chronic Intestinal Failure in Children: An International Multicenter Cross-Sectional Survey

**DOI:** 10.3390/nu14091889

**Published:** 2022-04-30

**Authors:** Antonella Lezo, Antonella Diamanti, Evelyne M. Marinier, Merit Tabbers, Anat Guz-Mark, Paolo Gandullia, Maria I. Spagnuolo, Sue Protheroe, Noel Peretti, Laura Merras-Salmio, Jessie M. Hulst, Sanja Kolaček, Looi C. Ee, Joanna Lawrence, Jonathan Hind, Lorenzo D’Antiga, Giovanna Verlato, Ieva Pukite, Grazia Di Leo, Tim Vanuytsel, Maryana K. Doitchinova-Simeonova, Lars Ellegard, Luisa Masconale, María Maíz-Jiménez, Sheldon C. Cooper, Giorgia Brillanti, Elena Nardi, Anna S. Sasdelli, Simon Lal, Loris Pironi

**Affiliations:** 1Department of Clinical Nutrition, OIRM-S, Anna Hospital, Città della Salute e della Scienza, 10126 Turin, Italy; alezodott@gmail.com; 2Ospedale Bambin Gesù, 00165 Rome, Italy; antonella.diamanti@opbg.net; 3Service des Maladies Digestives et Respiratoires de l’Enfant, Centre de Référence des Maladies Digestives Rares, Hôpital R Debré, 75019 Paris, France; marinier21@wanadoo.fr; 4Emma Children’s Hospital/Amsterdam University Medical Centers, 1105 AZ Amsterdam, The Netherlands; m.m.tabbers@amc.nl; 5Institute of Gastroenterology, Nutrition and Liver Diseases, Schneider Children’s Medical Center of Israel, Petach Tikva 4920235, Israel; anatguz@gmail.com; 6Sackler Faculty of Medicine, Tel-Aviv University, Tel-Aviv 6997801, Israel; 7Pediatric Gastroenterology and Endoscopy, IRCCS G, Gaslini Institute, 16147 Genoa, Italy; paologandullia@gaslini.org; 8Section of Paediatrics, Department of Translational Medical Science, University of Naples Federico II, 80138 Naples, Italy; mispagnu@unina.it; 9Department of Gastroenterology and Nutrition, Birmingham Children’s Hospital NHS Trust, Birmingham B4 6NH, UK; sue.protheroe@nhs.uk; 10Paediatric Hospital “Femme Mère Enfant de Lyon”, 69677 Lyon, France; noel.peretti@chu-lyon.fr; 11Pediatric Gastroenterology Unit, Helsinki University Hospital, Children’s Hospital Helsinki, 00290 Helsinki, Finland; laura.merras-salmio@hus.fi; 12Erasmus Medical Center, Sophia Children’s Hospital, 3015 CN Rotterdam, The Netherlands; jessie.hulst@sickkids.ca; 13Children’s Hospital Zagreb, Zagreb Medical University, 10000 Zagreb, Croatia; sanja.kolacek@gmail.com; 14Queensland Children’s Hospital, Brisbane, QLD 4101, Australia; looi.ee@health.qld.gov.au; 15Royal Children’s Hospital, Parkville, VIC 3052, Australia; joanna.lawrence@rch.org.au; 16Paediatric Liver, GI and Nutrition Centre, King’s College Hospital, London SE5 9RS, UK; jhind@nhs.net; 17Paediatric Hepatology, Gastroenterology and Transplantation, “Papa Giovanni XXIII” Hospital, 24127 Bergamo, Italy; ldantiga@asst-pg23.it; 18Paediatric Nutrition Service-Neonatal Intensive Care Unit, University Hospital of Padova, 35128 Padova, Italy; giovanna.verlato@aopd.veneto.it; 19University Children Hospital, LV-1004 Riga, Latvia; ieva.pukite@bkus.lv; 20Pediatric Department, University of Trieste, IRCCS Burlo Garofolo, 34137 Trieste, Italy; grazia.dileo@burlo.trieste.it; 21University Hospital Leuven, Leuven Intestinal Failure and Transplantation (LIFT), 3000 Leuven, Belgium; tim.vanuytsel@uzleuven.be; 22Bulgarian Association of Patients with Malnutrition, 1233 Sofia, Bulgaria; merryons@gmail.com; 23Sahlgrenska University Hospital, Institute of Medicine, Departement of Internal Medicine and Clinical Nutrition, University of Gothenburg, 41390 Gothenburg, Sweden; lasse.ellegard@nutrition.gu.se; 24Ospedale Orlandi, 37012 Bussolengo, Italy; luisa.masconale@aulss9.veneto.it; 25Department of Endocrinology and Nutrition, Hospital 12 de Octubre, 28041 Madrid, Spain; mariamaizj@gmail.com; 26University Hospitals Birmingham NHS Foundation Trust, Birmingham B15 2GW, UK; sheldon.cooper@nhs.net; 27Department of Medical and Surgical Sciences, Alma Mater Studiorum-University of Bologna, 40138 Bologna, Italy; giorgia.brillanti2@unibo.it; 28Department of Experimental, Diagnostic and Specialty Medicine, University of Bologna, 40138 Bologna, Italy; elena.nardi2@unibo.it; 29Clinical Nutrition and Metabolism Unit, Center for Chronic Intestinal Failure, IRCCS Azienda Ospedaliero-Universitaria di Bologna, 40138 Bolohna, Italy; annasimona.sasdelli@aosp.bo.it; 30Intestinal Failure Unit, Salford Royal Foundation Trust, Salford M6 8HD, UK; simon.lal@nca.nhs.uk

**Keywords:** children, chronic intestinal failure, home parenteral nutrition, body growth, intravenous supplementation, intestinal transplantation, transition

## Abstract

Background: The European Society for Clinical Nutrition and Metabolism database for chronic intestinal failure (CIF) was analyzed to investigate factors associated with nutritional status and the intravenous supplementation (IVS) dependency in children. Methods: Data collected: demographics, CIF mechanism, home parenteral nutrition program, z-scores of weight-for-age (WFA), length or height-for-age (LFA/HFA), and body mass index-for-age (BMI-FA). IVS dependency was calculated as the ratio of daily total IVS energy over estimated resting energy expenditure (%IVSE/REE). Results: Five hundred and fifty-eight patients were included, 57.2% of whom were male. CIF mechanisms at age 1–4 and 14–18 years, respectively: SBS 63.3%, 37.9%; dysmotility or mucosal disease: 36.7%, 62.1%. One-third had WFA and/or LFA/HFA z-scores < −2. One-third had %IVSE/REE > 125%. Multivariate analysis showed that mechanism of CIF was associated with WFA and/or LFA/HFA z-scores (negatively with mucosal disease) and %IVSE/REE (higher for dysmotility and lower in SBS with colon in continuity), while z-scores were negatively associated with %IVSE/REE. Conclusions: The main mechanism of CIF at young age was short bowel syndrome (SBS), whereas most patients facing adulthood had intestinal dysmotility or mucosal disease. One-third were underweight or stunted and had high IVS dependency. Considering that IVS dependency was associated with both CIF mechanisms and nutritional status, IVS dependency is suggested as a potential marker for CIF severity in children.

## 1. Introduction

Chronic intestinal failure (CIF) is defined as “the persistent reduction of the gut function below the minimum necessary for the absorption of macronutrients and/or water and electrolytes, such that intravenous supplementation (IVS) is required to maintain health and/or growth” [1,2].

In both adults and children, the most common pathogenetic mechanisms of CIF are short bowel syndrome (SBS), small bowel mucosal disease, and intestinal dysmotility, originating from either congenital, acquired, gastrointestinal, or systemic disorders [2,3]. CIF is the rarest organ failure, with a prevalence ranging from 20 to 80 cases per million adults [2] and from 14.1 to 56 cases per million children [4,5,6,7,8,9,10]. The primary treatment for CIF is IVS provided through home parenteral nutrition (HPN) programs [1,2]. Intestinal transplantation (ITx) is indicated for patients with irreversible CIF who are at risk of death because of CIF/HPN complications and for carefully selected patients experiencing a very poor quality of life [11].

The clinical feature and the outcome of both adults and children with CIF have been reported by several individual center and country surveys, which described factors associated with the probability of intestinal rehabilitation as well as with the risk of CIF/HPN-related major complications and death [10,12,13,14,15,16,17,18,19,20].

In 2015, the European Society for Clinical Nutrition and Metabolism (ESPEN) started a project based on an international database for CIF (ESPEN CIF Action Day database) and aimed at identifying a simple indicator of the severity of CIF [21]. In adults, it was shown that both IVS type and volume could be markers to categorize the severity of CIF because they were independently associated with the one-year odds of weaning from HPN, patients’ death, occurrence of cholestasis or liver failure due to intestinal failure-associated liver disease (IFALD), and central venous access device-related bloodstream infections [22].

In 2016, a number of pediatric CIF centers also started to contribute to the ESPEN CIF Action Day database. The present analysis of the international multicenter cohort of pediatric patients, at their first inclusion in the ESPEN CIF database between 2015 and 2020, aimed to describe the clinical features of children with CIF and the characteristics of the HPN programs and to investigate any associations between nutritional status parameters and the IVS dependency, defined by the ratio of daily total IVS energy over estimated resting energy expenditure.

## 2. Materials and Methods

This was an international, multicenter, cross-sectional, observational study on features of children with CIF at their first enrollment in the ESPEN database for CIF.

### 2.1. Participating Center Recruitment

In 2015, invitation to the HPN/CIF Centers to participate in the study occurred via representatives of the national Parenteral and Enteral Nutrition Societies of the ESPEN Council. In the following years, the invitation was made via an ESPEN newsletter to all ESPEN members [21]. The participation of pediatric centers commenced in 2016. Clinical units expressing an interest in participating were then sent the protocol study, the database, and instructions for data collection by the study coordinator (L.P.).

### 2.2. Patient Inclusion Criteria

HPN/CIF centers were required to enroll all patients who were dependent on HPN for CIF on 1 March of each year of data collection. Either patients with benign or malignant disease were admitted into the database. For the present study, only patients aged ≤ 18 years and with CIF due to benign disease were analyzed.

### 2.3. Data Collection and Schedule

Data were collected into a structured questionnaire embedded in an Excel (Microsoft Co., 2013 Washington US) database. The following items were gathered: age and sex; body weight, body length (children ≤ 2 year), or height (children > 2 years); underlying disease; pathophysiological mechanism of CIF; characteristics of the HPN program, including duration, number of days of IVS infusion per week, and type, volume, and total energy (from amino acids, glucose and lipids) of the IVS admixture. The pathophysiological mechanisms of CIF were recorded as SBS, intestinal dysmotility, mucosal disease, entero-cutaneous fistulas, and intestinal mechanical obstruction; SBS was further sub-categorized into end-jejunostomy or ileostomy (SBS-J or SBS type-1), jejuno-colonic anastomosis with part of the colon in continuity (SBS-JC or SBS type-2), and jejuno-ileal anastomosis with ileo-cecal valve and the entire colon in continuity (SBS-JIC or SBS type-3) [2,21,22]. The Pediatric Intestinal Pseudo-Obstruction (PIPO) recommendations from the European Society of Pediatric Gastroenterology, Hepatology, and Nutrition (ESPGHAN) were used to classify patients with intestinal dysmotility [23]. The information about the time (number of hours per day and timeframe within 24 h) used to provide IVS for the individual patient was requested from the participating centers subsequently because these data were not collected via the database.

### 2.4. Ethical Statement

The research was based on anonymized information taken from patient records at time of data collection. The study was conducted with full regard to confidentiality of the individual patient. Ethical committee approval was obtained by the individual contributing centers according to local regulations. The collected data were used only for the study purpose. Contributing centers have been anonymized for data analysis and presentation.

### 2.5. Intestinal Failure, Intravenous Supplementation, Nutritional Status Assessment, and Categorization

According to the standard classification for pediatric patients [5,6,7,8,9,10], the pathophysiological mechanisms of CIF were grouped into SBS (also including patients with enterocutaneous fistulas), intestinal dysmotility (also including patients with mechanical intestinal obstruction), and mucosal disease.

The daily mean volume and energy of IVS were calculated as follows: daily total volume (mL/day) or energy (kcal/day) = amount per day of infusion x number of infusions per week/7 [21]. The daily IVS total volume was calculated as the volume of the PN admixture and the fluids and electrolytes (FE).

The ESPEN database collected the total energy provided with the IVS but not the non-protein energy alone and not the energy intake via oral or enteral route. Therefore, IVS dependency was categorized by the ratio of daily IVS total energy over estimated resting energy expenditure (%IVSE/REE), calculated using Schofield equations [24].

The child’s nutritional status was assessed using z-scores for weight-for-age (WFA) and length-for-age (LFA, for patients aged 0–2 years) or height-for-age (HFA, for patients aged > 2 years) and body mass index-for-age (BMI-FA), calculated according to the World Health Organization Child Growth Standard (for patients aged 0–2 years) or to Centers for Disease Control Growth Charts 2000 (for patients aged > 2 years) [25]. BMI was calculated by Quetelet’s formula (weight (kg)/height (m^2^)). The normal z-scores range was considered between −1 and 1. z-scores < −2 for WFA, LFA/HFA, and BMI-FA were used to define underweight, stunting, and wasting, respectively [26], whereas BMI-FA z-score between 1 and 2 and >2 were considered as overweight and obese, respectively. The database did not collect data on prematurity. To determine if the lack of this parameter impacted on our findings, necrotizing enterocolitis (NEC), which predominantly occurs in preterm neonates, was used as a proxy for prematurity, and the mean z-scores for WFA, LFA/HFA, and BMI-FA were recalculated after excluding patients with NEC aged < 2 years.

### 2.6. Statistical Analysis

Continuous variables are reported as median and interquartile range (IQR) and categorical variables as absolute and relative frequencies. The Mann–Whitney’s and Spearman’s tests were used for univariate analysis of binary variables and of continuous and ordinal variables, respectively. Continuous variables were also categorized into intervals and the relative effects between the resulting ordinal variables were compared using the nonparametric multiple comparisons for relative effects test with Tukey’s test for the contrasts. The categorical variable “mechanism of CIF” was dichotomized and each dummy variable was analyzed individually. Pearson chi-square test, Fisher’s exact test, and linear-by-linear association test were used to analyze frequencies where appropriate.

The linear regression model was used for the multivariate analyses to investigate factors independently associated with z-scores of the nutritional status variables and %IVSE/REE as dependent continuous variables. The quantitative continuous variables and the categories of “mechanism of CIF”, which reached *p* < 0.1 on univariable analysis, were included as independent variables. When needed, the interaction between quantitative continuous variables, as a simple product of the variables in question, was included in the multivariate analysis as an independent variable.

Two-tailed *p*-values less than 0.05 were considered as statistically significant. The analyses were performed using the IBM SSPS Statistics package for Windows, version 27 (BM Co., Armonk, NY, USA) and R statistical software version 4.1.1., (Vienna, Austria), Statistical Package for the Social Sciences 20.0., (Chicago, IL, USA).

## 3. Results

Between 2015 and 2020, 558 children were enrolled, 13.8% of whom were included by 30 centers that primarily looked after adults and 86.8% by 19 dedicated pediatric centers from 18 countries (Appendix A). Most patients (55.4%) were included in 2016, 2.1% were included in 2015 (by adult centers), 14.6% in 2017, 10.0% in 2018, 7.0% in 2019, and 10.9% in 2020.

### 3.1. Characteristics and Clinical Features of the Patient Cohort

In the whole cohort, 57.2% were male, 70.1% < 10 years of age, 57% started HPN in the first year of life, and 54.5% were on HPN for ≤ 2 years at the time of the study. Three-quarters of the patients received the IVS infusion daily. The %IVSE/REE was >100% in 57.7% (>125% in 36.1%). All the patients were on cyclic overnight IVS infusion for 12–16 h. The main underlying diseases were PIPO (24.9%), congenital intestinal malformations (15.9%), congenital mucosal diseases (13.0%), and NEC (10.1%). Underweight and stunting were observed in around one-third of cases (WFA z-score < −2, 27.5%; LFA/HFA z-score < −2, 30.2%), whereas wasting was seen in 15.1% of patients (BMI-FA < −2). (Table 1). The results of the nutritional status z-scores at group level recalculated after excluding the 33 patients with NEC aged < 2 years were similar to those observed in the entire cohort (Appendix A).

The pathophysiological mechanism of CIF was SBS in 50.6% of patients (including seven patients, 1.3%, with enterocutaneous fistulas), dysmotility in 27.7% (including eight patients, 1.5%) (with mechanical obstruction), and mucosal disease in 21.8% (Figure 1).

The frequency of the mechanisms of CIF significantly differed between sex and age categories. Males made up 62.6% of those with SBS (*p* = 0.013), 49.7% of those with dysmotility (*p* = 0.038), and 54.4% of those with mucosal disease (*p* = 0.521) categories. With increase in age group, the percentage of patients with dysmotility or mucosal disease significantly increased, while the percentage of patients with SBS decreased (Figure 2).

### 3.2. Factors Associated with Patients’ Nutritional Status Parameters

The results of the univariate analysis are shown in Figure 3 and are reported in Appendix A. WFA z-score was negatively associated with patients’ age and %IVSE/REE and was lower in patients with mucosal disease and higher in patients with the SBS-JC mechanism of CIF. LFA/HFA z-score was negatively associated with the duration of the HPN and %IVSE/REE and was lower in patients with mucosal disease and higher in patients with dysmotility or SBS-JC. BMI-FA z-score was negatively associated with patients’ age and positively with HPN duration and was higher in patients with dysmotility.

The multivariate analysis (Table 2) investigating factors independently associated with nutritional status parameters demonstrated that both WFA and LFA/HFA z-scores were negatively associated with %IVSE/REE and mucosal disease CIF mechanism and that WFA z-score was also negatively associated with patients’ age and LFA/HFA z-score with the duration of HPN. BMI-FA z-score was negatively associated with patients’ age and positively associated with the duration of HPN.

### 3.3. Factors Associated with the IVS Energy Requirements

The univariate analysis demonstrated that %IVSE/REE was negatively associated with patients’ age, either at inclusion in the study or at starting of HPN. There was also an association between IVS requirements and CIF mechanism, with higher %IVSE/REE in patients with dysmotility and lower %IVSE/REE in those with SBS-JC or SBS-JIC (Figure 4 and Appendix A).

The multivariate analysis confirmed the association between the IVS-dependency and the mechanisms of CIF and the age at starting HPN (Table 3).

## 4. Discussion

This international cross-sectional survey describes the clinical features of pediatric patients with CIF, provides information into the CIF mechanism of patients transitioning into adulthood, and investigates factors associated with patients’ nutritional status and IVS dependency. Most patients were males, were in the first decade of life, had an early onset of CIF (within the first year of age), and were on HPN for less than 2 years. In the whole group, PIPO, intestinal malformations, and congenital mucosal diseases were the most frequent underlying diseases leading to CIF, and SBS was the mechanism of CIF in one-half of cases. The male predominance and the high proportion of SBS observed in our cohort are similar to results reported by the most recent national surveys carried out in Italy [6], the UK [9], and France [10] but differ from results of the 2015 ESPEN survey on CIF in adults, where most of the patients were females and SBS was the CIF mechanism in a higher proportion (66%) [21]. This difference between pediatric and adult patients with SBS can be explained by the fact that, in children, SBS prevails in males, and intestinal rehabilitation and weaning from HPN occurred in up to 80% [10,27], whereas in adults, SBS prevails in females, and intestinal rehabilitation occurred in up to 50% of cases [28,29] such that more adult females with SBS seem to remain on long-term HPN. Furthermore, the percentage of HPN-dependent children with SBS decreased within older age groups, becoming a minority (37.9%) in the 14–18-year-old category. These findings are in keeping with the higher rate of intestinal rehabilitation and HPN weaning in SBS than in PIPO and congenital mucosal disease [9,10] and highlight that most patients transitioning from pediatric to adult CIF centers are affected by these two very challenging mechanisms of CIF [30,31,32,33,34].

In agreement with previous reports [10,35,36,37], one-third of patients were either underweight or stunted, represented by WFA and/or LFA/HFA z-score < −2, and one-third had an IVS energy requirement > 125% of the REE, indicating a high dependency on HPN. Multivariate analysis demonstrated that nutritional status parameter z-scores were negatively associated with having mucosal disease as a mechanism of CIF, as previously observed [35,36], and with IVS dependency. Furthermore, the greater IVS dependency noted in patients with intestinal dysmotility and the lower dependency seen in those with SBS and colon in continuity (SBS types 2 and 3) are also very much in keeping with the clinical observation that oral/enteral feeding is typically very restricted in the former group [23] but preserved in the latter group of patients who also benefit from additional nutrient/fluid absorption from the colon [28,29]. In addition, the negative association between IVS dependency and patients’ age at starting HPN also indicates a more severe degree of intestinal failure and/or a greater difficulty in oral/enteral feeding when CIF arises at a very young age.

The pathogenesis of malnutrition in patients with CIF is multifactorial, including the degree of CIF, nutritional care provided, as well as factors such as inflammation related to the underlying disease, medications, medical procedures, and associated complications [38]. Nonetheless, the association between IVS dependency—which was expressed as ratio of daily total IVS energy over estimated REE (%IVSE/REE)—and CIF mechanism and the observation that nutritional parameters z-scores were lower in patients with greater IVS dependency while also being related to CIF mechanism, suggest that IVS dependency could be considered as a criterion to categorize of CIF severity that reflects both CIF mechanism and the relative risk of malnutrition. Indeed, the severity of CIF may cause decreased oral/enteral feeding and impaired growth, with both conditions requiring increased IVS of energy in order to meet patient needs as well as to stimulate growth. The use of the ratio of daily total IVS energy over estimated REE as a proxy for IVS dependency may be useful in clinical practice compared to other proxies that are currently used, such as the ratio of parenteral energy over total energy intake (oral nutrition, EN, and PN) since oral/enteral intake can be difficult to quantify especially in older children and since stool losses are not considered. The parenteral nutrition dependency index (PNDI), which is the ratio of non-protein energy intake (NPEI) from PN to resting energy expenditure (REE) [39,40], is similar to the ratio used in our study but excludes energy derived from amino acids. Since amino acids are either used for protein synthesis or undergo oxidation, it is clearly difficult to quantify the relative contribution of IV-supplied amino acids to these metabolic pathways when patients are also receiving oral and/or enteral feeding.

The large cohort of patients and the international and multicenter characteristics of this study are key strengths. The voluntary participation of the centers and the lack of parameters such as prematurity, oral nutritional intake, and overall protein intake as well as number of other factors (i.e., type of lipid emulsion) that may impact children’s growth that are not collected by the ESPEN CIF database may be considered weaknesses. In addition, inclusion of patients in a 5-year period (one-half during the first year of data collection and one-half in the remaining years) could be a matter of discussion because of potential variations in clinical practice that could have occurred in this time frame. Voluntary participation means the cohort may not be fully representative of the pediatric CIF population in each country. On the other hand, the variation in center size represents real-world practice [41] and is an adjunct value that will allow prospective comparison studies. Nutritional status parameters are commonly corrected in patients born prematurely up to 2 years of age when calculating z-scores. Thus, in order to determine if any lack of data on prematurity impacted our findings, we used NEC (which predominantly occurs in preterm neonates) as a proxy for prematurity and recalculated nutritional parameter z-scores after excluding 33 patients with NEC aged < 2 years; the results were similar to those observed in the entire cohort, suggesting—by inference—that prematurity may not have impacted our findings. While knowledge of oral/enteral intake and intestinal absorption may have allowed a more precise assessment of the degree of intestinal failure, our data suggest that IVS dependency could be a marker of CIF severity. It would be useful to validate this key finding through further prospective studies, taking into account that a number of other factors may have potentially impacted on nutritional status, such as HPN/CIF complications and treatments affecting growth (such as corticosteroids), as well as growth potential based on parental height, which were not collected in the present study [32].

## 5. Conclusions

We report a very large cohort of 558 pediatric CIF patients, with SBS being the main CIF mechanism at a young age, while intestinal dysmotility or mucosal disease were the predominant pathophysiological mechanisms in patients transitioning to adulthood. Furthermore, undernutrition and stunting were present in one-third of patients, and nutritional status as well as the degree of IVS energy dependency were mainly associated with CIF mechanism, while nutritional parameters were also negatively associated with IVS energy requirement. We therefore hypothesize that IVS dependency—expressed as the ratio of daily total IVS energy over estimated resting energy expenditure—could be a potential practical marker for CIF severity in pediatric patients. Prospective studies are required to validate this hypothesis.

## Figures and Tables

**Figure 1 nutrients-14-01889-f001:**
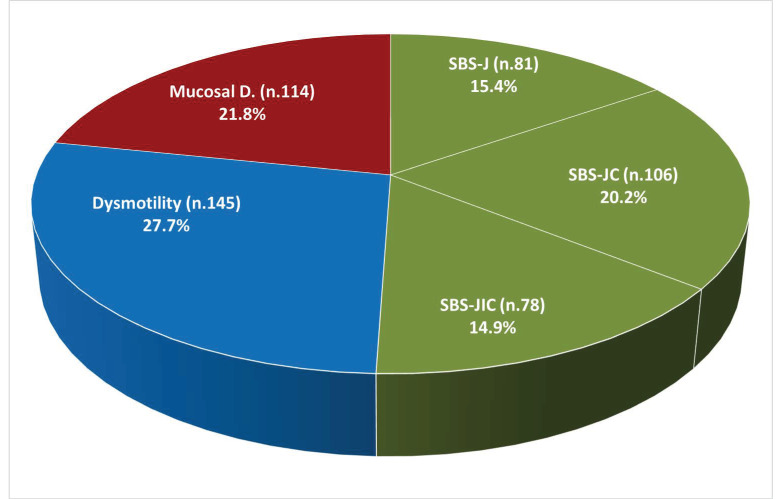
Pathophysiological mechanisms of chronic intestinal failure (n. 524). Abbreviations: SBS, short bowel syndrome; SBS-J, SBS with end jejunostomy or ileostomy (also included enterocutaneous fistulas, n. 7); SBS-JC, jejuno-colic anastomosis; SBS-JIC, jejunoileal anastomosis with an intact colon and ileocecal valve; Dysm, dysmotility (also included patients with mechanical obstruction, n. 8); Mucosal D, mucosal disease.

**Figure 2 nutrients-14-01889-f002:**
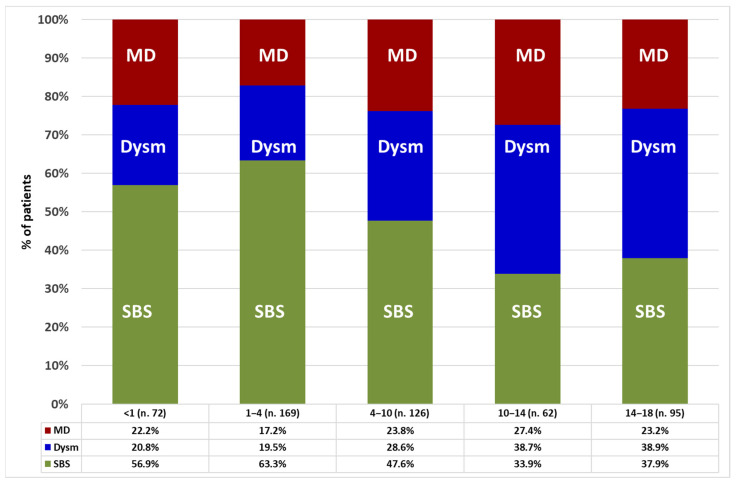
Prevalence of intestinal failure mechanisms in the patients’ age categories (years). Statistic: Pearson chi-square test for the whole group (*p* = 0.004); linear-by-linear association test for SBS (*p* < 0.001), dysmotility (*p* < 0.001), and mucosal disease (*p* = 0.282). Abbreviations: SBS, short bowel syndrome; Dysm, dysmotility; MD, mucosal disease.

**Figure 3 nutrients-14-01889-f003:**
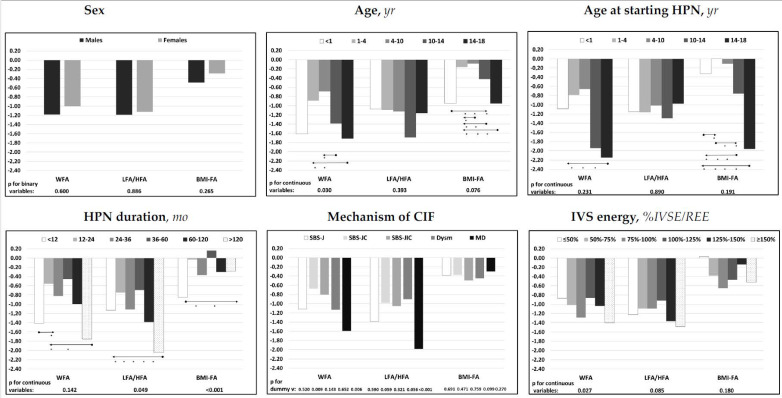
Associations of median z-scores for age of the nutritional status parameters with patients’ characteristics and with intravenous supplementation energy. Statistic: Spearman’s test was used for the continuous variables; Mann–Whitney’s test for the binary variable sex; the categorical variable “mechanism of CIF” was dichotomized, and each dummy variable was analyzed individually. The relative effects between the categories of the ordinal variables were compared using the nonparametric multiple comparisons for relative effects test with Tukey’s test for the contrasts. * *p* < 0.05 between categories; direction of the comparison. Abbreviations: WFA, body weight for age; LFA/HFA, body length for age (patients aged 0–2 years) or height for age (patients aged > 2 years); BMI-FA, body mass index for age; IVS energy (%IVSE/REE)*,* ratio of daily intravenous supplementation total energy over estimated resting energy expenditure; HPN, home parenteral nutrition; CIF, chronic intestinal failure; SBS-J, SBS with end jejunostomy or ileostomy (also included enterocutaneous fistulas, n. 7); SBS-JC, jejuno-colic anastomosis; SBS-JIC, jejunoileal anastomosis with an intact colon and ileocecal valve; Dysm, dysmotility (also included patients with mechanical obstruction, n. 8); MD, mucosal disease.

**Figure 4 nutrients-14-01889-f004:**
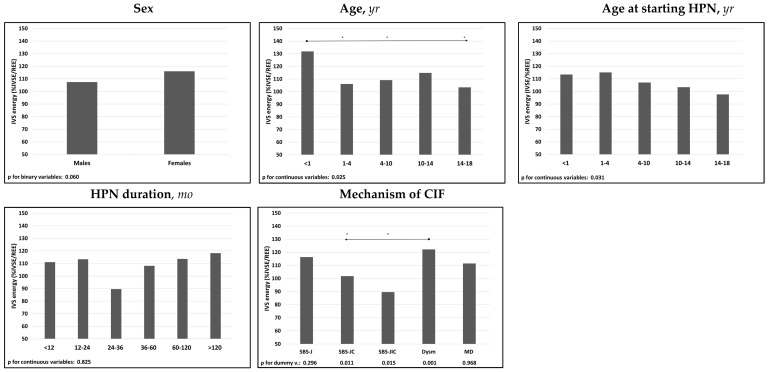
Associations of median IVS energy (%IVSE/REE) with patients’ characteristics. Spearman’s test was used for the continuous variables; Mann-Whitney’s test for the binary variable sex; The categorical variable “mechanism of CIF” was dichotomized and each dummy variable was analyzed individually; The relative effects between the categories of the ordinal variables were compared using the nonparametric multiple comparisons for relative effects test with Tukey’s test for the contrasts. * *p* < 0.05 between categories; direction of the comparison. Abbreviations: IVS energy (%IVSE/REE), ratio of daily intravenous supplementation total energy over estimated resting energy expenditure; HPN, home parenteral nutrition; CIF, chronic intestinal failure; SBS-J, SBS with end jejunostomy or ileostomy (also included enterocutaneous fistulas, n. 7); SBS-JC, jejuno-colic anastomosis; SBS-JIC, jejunoileal anastomosis with an intact colon and ileocecal valve; Dysm, dysmotility (also included patients with mechanical obstruction, n. 8); MD, mucosal disease.

**Table 1 nutrients-14-01889-t001:** Patient characteristics and clinical features.

**Sex** (n. 558)	
Males	57.2%
Females	42.8%
**Age,** year (n. 558)	
Median (IQR)	4.7 (9.7)
Category	
<1	13.6%
1 ≤ 4	32.3%
4 ≤ 10	24.2%
10 ≤ 14	12.0%
14–18	17.9%
**Age at starting HPN**, year (n. 458)	
Median (IQR)	0.7 (3.1)
Category	
<1	57.0%
1 ≤ 4	19.0%
4 ≤ 10	10.0%
10 ≤ 14	6.3%
14–18	7.6%
**HPN duration**, months (n. 541)	
Median (IQR)	18.0 (55.0)
Category	
<12	41.0%
12 ≤ 24	13.5%
24 ≤ 36	8.7%
36 ≤ 60	11.5%
60–120	15.9%
>120	9.4%
**IVS days/week**. (n. 548)	
Median (IQR)	7.0 (0.0)
Category	
2	1.1%
3–4	7.7%
5–6	14.2%
7	77.0%
**IVS energy (%IVSE/REE)** (n. 532)	
Median (IQR)	110.5 (61.8)
Category	
≤50%	11.7%
50 ≤ 75%	14.5%
75 ≤ 100%	16.2%
100–125%	21.6%
125 ≤ 150%	20.3%
≥150%	15.8%
**WFA z-score** (n. 545)	
Median (IQR)	−1.1 (2.0)
Category	
<−2	27.5%
−2 ≤ −1	25.1%
−1 ≤ 1	41.3%
1–2	4.6%
>2	1.5%
**LFA/HFA z-score**, (n. 523)	
Median (IQR)	−1.1 (2.1)
Category	
<−2	30.2%
−2 ≤ −1	23.7%
−1 ≤ 1	39.0%
1–2	4.0%
>2	3.1%
**BMI-FA z-score**, (n. 522)	
Median (IQR)	−0.3 (12.7)
Category	
<−2	15.1%
−2 ≤ −1	19.2%
−1 ≤ 1	52.1%
1–2	10.7%
>2	2.9%
**Underlying disease** (n. 547)	
PIPO *	24.9%
Intestinal malformation ^	15.9%
Volvulus	13.0%
Congenital mucosal disease ^#^	13.0%
Necrotizing enterocolitis	10.1%
Mesenteric ischemia	3.5%
Crohn’s disease	2.6%
Neurologic disease	1.6%
Autoimmune enteropathy	1.6%
Pancreatic disease	1.1%
Others	12.8%

Abbreviations: WFA, weight-for-age; LFA/HFA, length-for-age (patients aged 0–2 years) or height-for-age (patients aged > 2 years); BMI-FA, body mass index-for-age; IVS energy (%IVSE/REE)*,* ratio of daily intravenous supplementation total energy over estimated resting energy expenditure; * PIPO: pediatric intestinal pseudo-obstruction: idiopathic 95, Hirschsprung’s disease 27, others 8, not specified 6. ^ Intestinal malformation: atresia (esophageal or intestinal) 17, gastroschisis 8, laparoschisis 4, apple peel syndrome 5, others 3, not specified 51. ^#^ Congenital mucosal disease: microvillus inclusion disease 22, tufting enteropathy 20, trichohepatoenteric syndrome 14, congenital diarrhea 6, others 5, not specified 4. Others: mitochondrial disorder 5, intestinal lymphangectasia 5, protein-losing enteropathy 4, common variable immunodeficiency 4, surgical complications 4, adhesions 3, trauma 2, infectious disease 2, collagenous vascular disease 2, others 39.

**Table 2 nutrients-14-01889-t002:** Multivariate analysis of factors associated with nutritional status parameters (z-scores).

Dependent Variable	Variables Entered	Predictor Variables	B	*p*
WFA z-score	Age IVS energy (%IVSE/REE) SBS-JIC MD	Constant Age IVS energy (%IVSE/REE) MD	−0.384 −0.032 −0.006 −0.330	0.072 0.018 <0.001 0.079
LFA/HFA z-score	Duration of HPN IVS energy (%IVS/REE) SBS-JIC SBS-JC MD	Constant Duration of HPN IVS energy (%IVSE/REE) MD	−0.417 −0.004 −0.622 −0.005	0.069 0.030 0.005 0.005
BMI-FA z-score	Age Duration HPN Dysm Interaction term between age and duration of HPN	Constant Age Duration HPN	−0.384 −0.079 0.008	0.001 <0.001 <0.001

Statistic: the dependent variables z-scores were included as continuous variables. The independent quantitative variables were included as continuous. The categories of the independent variable mechanism of CIF were included as dummy variables. Abbreviations: WFA, body weight for age; LFA/HFA, body length for age (patients aged 0–2 years) or height for age (patients aged >2 years); BMI-FA, body mass index for age; IVS energy (%IVSE/REE)*,* ratio of daily intravenous supplementation total energy over estimated resting energy expenditure; HPN, home parenteral nutrition; CIF, chronic intestinal failure; SBS-JC, short bowel syndrome with jejuno-colic anastomosis; SBS-JIC, short bowel syndrome with jejunoileal anastomosis with an intact colon and ileocecal valve; Dysm, dysmotility; MD, mucosal disease.

**Table 3 nutrients-14-01889-t003:** Multivariate analysis of factors associated with the IVS energy dependency (%IVSE/REE).

Dependent Variable	Variables Entered	Predictor Variables	B	*p*
%IVSE/REE	Sex Age Age at starting HPN Dysm SBS-JC SBS-JIC	Constant Age at starting HPN Dysm SBS-JC SBS-JIC	118.238 −1.153 11.967 −15.259 −18.491	0.000 0.024 0.059 0.025 0.012

Statistic: The dependent variable %IVSE/REE was included as continuous variable. The independent quantitative variables were included as continuous. The categories of the independent variable mechanism of CIF were included as dummy variables. Abbreviations: %IVSE/REE, ratio of daily intravenous supplementation total energy over estimated resting energy expenditure; HPN, home parenteral nutrition; SBS-JC, short bowel syndrome with jejuno-colic anastomosis; SBS-JIC, short bowel syndrome with jejunoileal anastomosis with an intact colon and ileocecal valve; Dysm, dysmotility (also included patients with mechanical obstruction, n. 8).

## Data Availability

Restrictions apply to the availability of these data. Data were obtained from each contributing center and are available from the corresponding author with the permission of the coordinators of the contributing centers.

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
