# Peer review of "Chronic Intestinal Failure in Children: An International Multicenter Cross-Sectional Survey"

_nutrients, 2022, doi:10.3390/nu14091889_

Round 1
Reviewer 1 Report
The study represents the largest cohort of patients with IF, on Nutritional rehabilitation programs for the longest time, so far; this is a major strength.
No information is given on the time used for administration of the IVS or the HPN, whether all patients were on “cyclic” TPN at night or the number of hours that were used to give / receive the IV nutrients. This may be an important factor having an impact in the oral or enteral amount and type of delivery of extra nutrients mainly on those with Short Bowel syndrome and subsequent adaptation, which was definitely the group that apparently had the best chances for this advancement to occur. The least number of hours the IVS is given, the night vs. day administration, may have a major impact on alteration of circadian rhythms and digestive cycles which again may impact adaptation severely. Perhaps some comment if the information was or can be made available, would add strength to the study.
Was there any information on fluid requirements for hydration, besides the IVS or HPN given, to maintain fluid and electrolyte balances? A large number of patients in all categories or CIF may have increase fluid loses through the GI tract, which may also impact growth.
Perhaps some increased emphasis on the fact that making younger children grow better will enhance and improve the possibility of adaptation, may also be important to state.
Reviewer 2 Report
Dear Editor, Dear Authors,
Thank you for the opportunity to review your manuscript.
The work appears significant and essential study related to intestinal failure of IV-dependent children. The finding of this study provides important results and is original. However, some questions remain unanswered or need clarification.
Major comments:
Line 111: Even though you have collected the data on the type of HPN, the different compositions of home parenteral nutrition, especially lipid emulsion, used by various institutions or for different categories of patients may influence the results. For instance, some lipid emulsion causes more adverse effect than others. It would be included in the analysis to mitigate the bias over total energy or under limitations.
Figures 2 and 3: The categorizations of age should be clarified. As the subjects fall under 18, the grouping should be justified. For instance, the children are often grouped into less than 60 months, 5-<10, and adolescents to assess malnutrition. The regrouping may give a significant association, and it would be considered.
Lines 144-145: It would be better to include how normal/ overweight among children using z-score values was assessed in the methodology as the table depict the results.
Line 155: Tukey's test assumes that the data is normally distributed. Please specify or clarify the test that you used for non-parametric multiple comparisons.
Line 268: It would be better to include the positive Z score values in the bar figures. Difficult to comment on the figures as those figures are overlapped.
Line 90: Please define the IVS dependency here.
Line 173: It would be better to discuss the reason for more patients (55.4%) were included in 2016 alone to mitigate the variation in clinical practices over the years as updated guidelines.
Line 187; Table 1: Selected patients were CIF due to benign disease; it would be better to include the possible reasons for the occurrence of obesity.
Mino comments:
Line 43: It would be better to use "weight-for-age" and "length-for-age", including hyphens instead of "age of weight" and instead of "age of length" to maintain consistency.
Line 49: It would be better to include the causal association, whether positively or negatively, with WFA and/or %IVSE/REE.
Line 53-54: Please avoid the repetition as you mentioned the calculation for IVS-dependency in lines 44-45; hence it would be better to remove it.
Lines 61-62: Micronutrients, including vitamins and minerals/trace elements, are missing in the definition.
Line 47: It would be better to provide the full form of SBS.
Line 117-120: Please maintain the consistency of the font style.
Line 128: It would be better to use the appropriate subheading as some contents are unrelated to statistical analysis.
Line 139: Please remove the hyphen after [24].
Line 181: Please provide the abbreviation "NEC", as mentioned earlier.
Line 187: Table 1: It would be better to revise the age category as 1-<4; 4-<5, etc., and similar for other categories. For instance, in IVS energy, <50% and again 50-75%. Please revise to avoid double counting or missing counting.
Line 188 and 289: Please replace weight-for-age and length-for-age or height-for-age by adding hyphens.
Line 188: Please replace '0-£2 years' from 0-2 years.
Line 208: Please resize figure 1 and rearrange figure 2.
Line 199-200: A separation space between paras.
Lines 323-325: Inconsistency in font size.

Reviewer 3 Report
This was an international, multicenter, cross-sectional observational study on features of children with chronic intestinal failure at their first enrollment in the ESPEN database.
The study is very interesting, especially due to the high number of patients included. The use of the ratio of daily total intravenous supplementation (IVS) energy over estimated resting energy expenditure as a proxy for IVS -dependency is a good idea.
The work could be accepted, but the following suggestions should be clarified beforehand.
1 - The study of anthropometry in the form of z-scores is totally adequate. However, the exclusive use of the weight-for-age z-score and not the weight-for-height z-score (in the form of the weight/height ratio or BMI for all ages, as already recommended by the WHO [page vii; Guideline: assessing and managing children at primary health-care facilities to prevent overweight and obesity in the context of the double burden of malnutrition. Updates for the Integrated Management of Childhood Illness (IMCI). Geneva: World Health Organization; 2017. Licence: CC BY-NC-SA 3.0 IGO.]) leads to a loss of information regarding to the anthropometric impact of the disease in these children for the present and the future. This parameter should be added to the study.
2 - Were patients and/or their families asked for permission to use their data?
3 - The statistical analysis section shows (lines 129-149) the characteristics of certain variables, the way in which they have been obtained, their form of expression, and the reason for their choice. They should be written in a separate section.
4 - It remains to comment on the limitations the lack of certain data from the registries.
5 - In Table 1, in the WFA and LFA/HFA sections, z-score is missing before the number of available data. Below it is not necessary to put again z-score, median (IQR) and z-score category, %,
The large number of unspecified intestinal malformations is striking.
6 - In Figure 2, in abbreviations at the bottom, the abbreviation for mucosal disease is missing.
7 - Figure 3. I do not understand the meaning of the graphs in the upper part: sex, age and age at starting HPN.
Please clarify the statistics at the bottom.
8 - Check the bibliography so that all citations are written in the same way:
The same number of authors when there are more than six; the initial and final pages; the order in the writing of the year, volume and pages; add or not doi, PMID, PMCID.
Round 2
Reviewer 3 Report
The work has improved in terms of its structure, and ethical concerns about this study have been clarified. The paper could be accepted after the minor changes suggested below.
Line 148. Please, put de full form of FE and then the abbreviation.
Line 157. Please provide the abbreviation "BMI", as mentioned earlier.
Table 1. - In Table 1, in the BMI section, z-score is missing before the number of available data.
